# *TLR5* Variants Are Associated with the Risk for COPD and NSCLC Development, Better Overall Survival of the NSCLC Patients and Increased Chemosensitivity in the H1299 Cell Line

**DOI:** 10.3390/biomedicines10092240

**Published:** 2022-09-09

**Authors:** Jurica Baranašić, Maja Šutić, Calogerina Catalano, Gordana Drpa, Stefanie Huhn, Dragomira Majhen, Davor Nestić, Matea Kurtović, Lada Rumora, Martina Bosnar, Andrea Vukić Dugac, Irena Sokolović, Sanja Popovic-Grle, Nada Oršolić, Sanda Škrinjarić-Cincar, Marko Jakopović, Miroslav Samaržija, Alexander N. R. Weber, Asta Försti, Jelena Knežević

**Affiliations:** 1Laboratory for Advanced Genomics, Division of Molecular Medicine, Ruđer Bošković Institute, Bijenička Cesta 54, 10000 Zagreb, Croatia; 2Division of Molecular Genetic Epidemiology, DKFZ, 69120 Heidelberg, Germany; 3Clinical Department for Respiratory Diseases Jordanovac, School of Medicine, University Hospital Centre Zagreb, University of Zagreb, 10000 Zagreb, Croatia; 4Department of Internal Medicine V, Heidelberg University Hospital, University of Heidelberg, 69120 Heidelberg, Germany; 5Laboratory for Cell Biology and Signaling, Division of Molecular Biology, Ruđer Bošković Institute, 10000 Zagreb, Croatia; 6Department of Medical Biochemistry and Hematology, Faculty of Pharmacy and Biochemistry, University of Zagreb, 10000 Zagreb, Croatia; 7Fidelta d.o.o., Prilaz Baruna Filipovića 29, 10000 Zagreb, Croatia; 8Faculty of Science, Department of Biology, University of Zagreb, 10000 Zagreb, Croatia; 9Faculty of Medicine, J.J. Strossmayer University of Osijek, 31000 Osijek, Croatia; 10Department of Immunology, University of Tübingen, 72074 Tübingen, Germany; 11iFIT—Cluster of Excellence (EXC 2180) ‘Image-Guided and Functionally Instructed Tumor Therapies’, University Hospital Tübingen—Internal Medicine VIII, 72076 Tübingen, Germany; 12Hopp Children’s Cancer Center (KiTZ), 69120 Heidelberg, Germany; 13German Cancer Research Center (DKFZ), Division of Pediatric Neurooncology, German Cancer Consortium (DKTK), Im Neuenheimer Feld 580, 69120 Heidelberg, Germany; 14Faculty for Dental Medicine and Health, University of Osijek, 31000 Osijek, Croatia

**Keywords:** COPD, NSCLC, TLR5, SNP, survival, chemoresistance

## Abstract

Chronic obstructive pulmonary disease (COPD) is considered as the strongest independent risk factor for lung cancer (LC) development, suggesting an overlapping genetic background in both diseases. A common feature of both diseases is aberrant immunity in respiratory epithelia that is mainly regulated by Toll-like receptors (TLRs), key regulators of innate immunity. The function of the flagellin-sensing TLR5 in airway epithelia and pathophysiology of COPD and LC has remained elusive. We performed case–control genetic association and functional studies on the importance of *TLR5* in COPD and LC development, comparing Caucasian COPD/LC patients (*n* = 974) and healthy donors (*n* = 1283). Association analysis of three single nucleotide polymorphisms (SNPs) (rs725084, rs2072493_N592S, and rs5744174_F616L) indicated the minor allele of rs2072493_N592S to be associated with increased risk for COPD (OR = 4.41, *p* < 0.0001) and NSCLC (OR = 5.17, *p* < 0.0001) development and non-small cell LC risk in the presence of COPD (OR = 1.75, *p* = 0.0031). The presence of minor alleles (rs5744174 and rs725084) in a co-dominant model was associated with overall survival in squamous cell LC patients. Functional analysis indicated that overexpression of the rs2072493_N592S allele affected the activation of NF-κB and AP-1, which could be attributed to impaired phosphorylation of p38 and ERK. Overexpression of TLR5^N592S^ was associated with increased chemosensitivity in the H1299 cell line. Finally, genome-wide transcriptomic analysis on WI-38 and H1299 cells overexpressing TLR5^WT^ or TLR5^N592S^, respectively, indicated the existence of different transcription profiles affecting several cellular pathways potentially associated with a dysregulated immune response. Our results suggest that *TLR5* could be recognized as a potential biomarker for COPD and LC development with functional relevance.

## 1. Introduction

Chronic obstructive pulmonary disease (COPD) is a chronic inflammatory condition characterized by unfavorable lung remodeling contributing to lung cancer (LC) susceptibility [1]. Several epidemiological studies have identified an association between the presence of airflow obstruction and the incidence of LC [2]. Both diseases are predominantly associated with exposure to cigarette smoke, however it is well accepted that COPD is the largest independent risk factor for LC development, indicating the existence of overlapping pathogenic mechanism that triggers the development of both diseases [3]. In a comparative study among patients with newly diagnosed LC and matched healthy controls, the prevalence of COPD in individuals with and without LC was 50% and 8%, respectively [4]. Additionally, epidemiological studies have shown that tobacco smoke exposure accounts for nearly 80–90% of all COPD and LC cases, but only 10–15% of smokers develop LC while 20–30% develop clinically significant COPD [5]. Lung cancers are traditionally subdivided into two main histological types: small-cell lung cancers (SCLC) and non-small-cell lung cancers (NSCLCs). NSCLCs represent 80–85% of all lung cancers. NSCLCs are further classified into different subtypes where squamous-cell carcinoma (SQC) and adenocarcinomas (AdC) account for the majority of NSCLC cases. SQC accounts for approximately 20–30% of NSCLC cases and AdCs account for about 40–50% of NSCLC cases. Because many lung cancer cases are presented in advanced stages, most patients are unresectable and the 5-year survival rates for many subtypes fall to below 2% for SCC [6]. Therefore, there is an unmet need for a constant search for new diagnostic and prognostic molecular biomarkers. 

It is well accepted that innate immunity and uncontrolled inflammation play an important role in the lung parenchyma remodeling and contribute to the development of the COPD and lung cancer. The innate immune system is based on pattern recognition receptors (PRRs) and their ligands. Binding of the specific ligand to the receptor results in the activation of complex signaling cascades triggering host-defense responses. Toll-like receptors (TLRs) are expressed in both tissue, including the human lung [7], and immune cells where they play an important role in innate immunity, causing inflammatory responses. However, TLR signaling is also an integral part of the homeostasis maintenance between damage and repair mechanisms [8]. Most of the TLRs are expressed in different tissues, and besides their contribution to innate immunity, they have a potential role in signaling the presence of microbiota, tissue destruction, chronic organ injury, differentiation, and neoplastic disease [9]. Real-time quantitative PCR showed that all TLR genes are expressed in human lung tissue [7]. The level of expression of TLRs is modulated by exogenous and endogenous stimuli, which allows host cells to adapt to changes in their environment [10]. TLRs are also expressed on the tumor cells and their role in the immune response to tumor cells has been confirmed. Their activation controls key signaling pathways for tumorigenesis and tumor progression [11]. They can be activated by pathogen-associated molecular patterns (PAMPs) or damage associated molecular patterns (DAMPs) [12]. Signaling cascade activated upon specific TLR ligand binding results in recruitment of adaptive molecules, triggering the activation of transcription factors, such as NF-κB, AP-1, or IRFs, and the production of inflammatory cytokines and other factors [13]. Upon activation of TLRs the production of inflammatory cytokines like TNF, IL-6, and IL-12; and type I interferons (IFNs), such as IFNα and IFNβ, are triggered [14]. Transcription factor NF-κB is activated by inflammatory mediators and oxidative stress and can be a link between inflammation and LC as it is activated in bronchial epithelium and inflammatory cells of respiratory tract of COPD patients and premalignant lesions of bronchial epithelium and neoplastic LC cells [15]. It is well known that inflammation stimulates carcinogenesis, while, at the same time, it triggers immune mechanisms that can suppress tumor growth [16]. Although TLR7, a sentinel of viral infection, has already been studied in the context of the lung cancer [17], little is known about the contribution of TLR5 to immune responses in the human lung in this context. TLR5 is receptor for flagellin, the promoter of the bacterial flagellum [18] and has been studied in the context of colorectal cancer. In our previous work, we showed that functional alleles correlated with survival—the frequent *TLR5* SNPs are associated with altered survival in a large cohort of Caucasian patients with colorectal cancer. Although rs5744174_F616L was associated with increased survival, rs2072493_N592S was associated with decreased survival due to higher responsiveness to flagellin [19]. As both the healthy and diseased lung is also exposed to flagellated bacteria, e.g., Pseudomonas aeruginosa, a key etiological agent for pulmonary infection [20], we sought to investigate an association between TLR5, COPD, and LC.

Based on our previously obtained results on colorectal cancer, we hypothesized that selected polymorphisms in *TLR5* gene could be associated with COPD and/or LC predisposition and potentially recognized as yet unknown biomarker of LC development in the patients diagnosed with COPD. The aim of present study was to assess the prevalence of selected SNPs located in the *TLR5* gene among COPD, LC, and healthy populations and analyze the impact of these SNPs on COPD and LC risk and clinical characteristics. We also aimed to investigate the risk of the lung cancer development in the context of existing COPD. By performing comprehensive in vitro studies, we further analyzed functional consequences of the rs2072493_N592S allele, an established functional *TLR5* germline variant. Results of our analysis show that tested *TLR5* functional variant (rs2072493_N592S), associated with increased risk for COPD, lung cancer, and lung cancer development in COPD patients, with functional relevance, indicating that these new insides of TLR5 function which might lead to better understanding of the TLR5 and flagellin role in COPD and lung cancer development.

## 2. Materials and Methods

### 2.1. Study Participants

All patients diagnosed with COPD or primary lung cancer, with or without COPD, and their clinical data were collected at Clinical Hospital Centers in Zagreb and Osijek, Croatia. Healthy donors were collected at Department of Transfusion Medicine, Zagreb, Croatia. This study was approved by institutional ethics committees and informed consents were signed by all participants. This study was performed in accordance with the Declaration of Helsinki. We enrolled 1283 healthy donors and 974 patients. Patients were divided into three groups: COPD_only_ (500, 51.33%), COPD + LC (280, 28.75%), and LC_only_ (194, 19.92%). For COPD patients, detailed assessment criteria are described elsewhere [21]. Lung cancer patients were included in the study after confirmation by histopathological diagnosis, according to the World Health Organization classification criteria [22]. Lung cancer patients has been further classified according to the TNM staging system which categorizes tumors on the basis of primary tumor characteristics (T), the presence or absence of regional lymph node involvement (N), and the presence or absence of distant metastases (M). They were enrolled in the study from October 2012 till March 2016. The control group of healthy volunteers was recruited during the regular blood donation process by the Department of Transfusion Medicine, Zagreb, and represents the general healthy population characterized by good basic health status. They were recruited from August 2015 till January 2016. Demographic and clinical data are presented in Table 1 and Table 2.

### 2.2. SNP Selection Criterion

The SNP selection criterion was based on the hypothesis that *TLR5* coding/regulatory variants could be associated with dysregulated receptor function, different cell responses, and increased chance of developing NSCLC in the presence of COPD by modulating activation and regulation of inflammatory microenvironment. We used the National Center for Biotechnology Information (NCBI) database available on PubMed (https://www.ncbi.nlm.nih.gov/snp/, accessed on 1 May 2017) and analyzed a list of polymorphisms within the functionally significant protein/gene regions. Altogether 3 SNPs located in the *TLR5* gene were analyzed; rs725084 (promotor/regulatory SNP), together with rs2072493_N592S and rs5744174_F616L located in the coding region of the gene. Based on NCBI, minor allele frequency for all tested SNPs was higher than 1%. Deleteriousness of the amino acid changes were predicted using SIFT (http://sift.jcvi.org/, accessed on 1 May 2017) and PolyPhen-2 (http://genetics.bwh.harvard.edu/pph2/, accessed on 1 May 2017). More detail on the selected SNPs is shown in Table 3.

### 2.3. DNA Sample Preparation and Genotyping

Genomic DNA analysis was performed on DNA isolated from peripheral blood by salting out procedure. KASP (LGC Genomics, Berlin, Germany) or TaqMan (Applied Biosystems, Waltham, MA, USA) allelic discrimination methods were used for genotyping analysis, according to the LGC Genomics’ and TaqMan’s PCR conditions. Samples were amplified in 384-well format using Hydrocycler 16 (LGC Genomics, Berlin, Germany). Genotype detection was carried out on the ViiA 7 Real-Time PCR System (Applied Biosystems, Waltham, MA, USA).

### 2.4. Statistical Analysis for Association Studies

The genetic association was estimated by unconditional logistic regression computing odds ratios (ORs), 95% confidence intervals (CIs), and *p*-values. All analyses were adjusted for age (at onset of COPD and at diagnosis of LC), gender, and smoking status. *p* values were considered significant at *p* ≤ 0.05. Standard deviation (SD) was used to describe the variation in the data values. Statistical analyses were performed using MedCalc version 15.8 (MedCalc Software, Ostend, Belgium) and SAS software version 9.2 (SAS Institute, Heidelberg, Germany). Unadjusted associations were evaluated by χ^2^ test. The effect of the different genotypes on survival were evaluated using the Kaplan–Meier method and were compared using log-rank testing. Follow-up time was calculated from the date of disease diagnosis to the death by any cause. Analysis of different parameters for prognostic significance was completed by univariate and multivariate Cox proportional hazard models. Correlation between TLR5 polymorphisms and clinical data was performed using Fisher’s exact test. *p* values were considered significant at *p* ≤ 0.05. Post hoc analysis was also performed using Fisher’s exact test, *p* ≤ 0.05 were considered significant. For functional analyses, data were analyzed using GraphPad Prism (GraphPad Software, Inc., San Diego, CA, USA). For the comparisons of wild type (WT) with their respective SNP variants, *p* values were determined using an unpaired *t* test or a Mann–Whitney test, as indicated. *p* < 0.05 was generally considered statistically significant.

### 2.5. Cells Cultures

Human lung fibroblast cell line WI-38 (ATCC^®^ CCL-75™), human non-small cell lung cancer cell line NCI-H1299 (ATCC^®^ CRL-5803™) and human embryonic kidney cell line HEK293 (ATCC^®^ CRL-1573) were obtained from ATCC. The WI-38 cell line was cultured in MEM medium supplemented with 10% fetal bovine serum (FBS), while the H1299 and HEK293 cell lines were cultured in DMEM medium supplemented with 10% FBS. All the cell lines were incubated at 37 °C in an atmosphere of 5% CO_2_.

### 2.6. Generation of Adenoviral Vectors Containing TLR5^WT^ and the TLR5^N592S^ as a Transgenes

In order to allow transient transfection of cells with TLR5^WT^ or TLR5^N592S^, adenoviral vector based on replication deficient adenovirus type 5 were prepared (Ad5-TLR5-WT and Ad5-TLR5-N592S). TLR5-WT-HA and TLR5-N592S-HA were excised from pcDNA3.1 described before [19,23], with PmeI restriction enzyme (New England Biolabs, Ipswich, MA, USA), and cloned into a linearized pShuttle-CMV plasmid from the AdEasy system (New England Biolabs, Ipswich, MA, USA), following dephosphorization by the Shrimp Alkaline Phosphatase enzyme (New England Biolabs, Ipswich, MA, USA). TLR5-WT-HA and TLR5-N592S-HA ligation mixtures were transformed into commercial Subcloning Efficiency^®^ DH5α ™ Chemically Competent Cells (Ref. 18265-017, Invitrogen by Thermo Fisher Scientific, Waltham, MA, USA). Subsequent sequencing analysis confirmed transformation in selected colonies. The pShuttle-CMV-TLR5WT and pShuttle-CMV-TLR5N592S vectors were linearized with a PmeI restriction enzyme and then electroporated into commercial bacteria BJ5183-Ad1 (BJ5183-AD-1 Electroporation Competent Cells, Agilent, Cat. No. 200157, Santa Clara, CA, USA). The transformation mixture was plated on LB-Kanamycin agar plates overnight. Recombination was confirmed by EcoRV restriction enzyme (New England Biolabs, Ipswich, MA, USA). Prior to transfection into HEK293 cells adenoviral constructs were linearized with PacI enzyme (New England Biolabs, Ipswich, MA, USA). HEK293 cells were seeded in 6-well plate and after 24 h transfected with Lipofectamine 2000 (Thermofisher, Waltham, MA, USA). Twelve days after transfection, cells were harvested and the viral vectors were liberated by three freeze/thaw cycles and 40 T75 flasks of HEK293 cells were infected with WT-pAd1 or N592S-pAd1 lysates. Two days after infection 50% of cells showed cytopathic effect (CPE). The cell pellet was left in about 4–6 mL of medium and then 3 freeze–thaw cycles were performed. Adenoviral vectors were purified by ultracentrifugation in cesium chloride (CsCl) density gradient. CsCl was removed from adenoviral vectors by using PD-10 desalting column (Sephadex G-25M, Amersha Pharmacia Biotech, Amersham, UK) in PBS according to the manufacturer’s protocol. Glycerol was added in final 10% (*v*/*v*) before freezing. The number of viral particles required for optimal cell infection was determined using the titration method in combination with Western blot for detection of protein expression of HA-tag directly linked to C-terminus of TLR5.

### 2.7. Reporter Gene Assays

To explore the activity of NF-κB and AP-1 signaling pathways, Cignal Reporter Assay Kits (Qiagen, Hilden, Germany) for NF-κB and AP-1 were used according to the manufacturer’s protocol. Briefly, 2 × 10^4^ H1299 or WI-38 cells were seeded (100 μL of medium, 96-well format). The following day cells were transfected with 200 ng of DNA using Lipofectamine 2000 (Thermofisher, Waltham, MA, USA). After 4–6 h the Lipofectamine containing medium was replaced with complete growth medium. Cells were incubated at 37 °C for another 24 h. The next day, cells were stimulated with flagellin (final concentration 50 ng/mL) for 24 h. After that medium was washed, cells were stored at −80 °C for 1 h and finally resuspended in Dual-Glo^®^ Subtrate 1 and Firefly luminescence was measured. Afterwards, Dual-Glo^®^ Stop & Glo^®^ Subtrate 2 was added and the Renilla luminescence was measured (Promega, Madison, WI, USA). Firefly to Renilla luciferase ratio was calculated. A mixture of a constitutively expressing GFP construct, constitutively expressing Firefly luciferase construct, and constitutively expressing Renilla luciferase construct (40:1:1) was used as a positive control.

### 2.8. Immunoblot Analysis

For the purpose of protein analysis, cell lines were plated in 12-well-plates, at a density of 10^5^ cells per well. The next day, the cells were infected with adenovirus 5 containing TLR5-wt and TLR5-N592S constructs, respectively. After two hours, the medium was removed and replaced with fresh medium. The next day, the medium was replaced with a medium containing 50 ng/mL flagellin (Invivogen, San Diego, CA, USA), while in control samples a medium without flagellin was added. After 15, 30, 60, and 120 min, the medium was removed and 200 μL of 1× hot lysis buffer was added to cells for protein extraction and resolved on 10% SDS-PAGE gel. The proteins were transferred onto a PVDF membrane and blocked in 5% fat milk. The following antibodies were used for immunoblotting: 1:500 iKBα (sc-371, Santa Cruz Santa Cruz Biotechnology, Dallas, TX, USA), phospho-p38 (sc-166182, Santa Cruz Santa Cruz Biotechnology, Dallas, TX, USA), 1:200 phospho-ERK (sc-7383, Santa Cruz Santa Cruz Biotechnology, Dallas, TX, USA), and 1:1 000 vinculin (sc-73614, Santa Cruz Santa Cruz Biotechnology, Dallas, TX, USA) as loading control; and 1: 10,000 anti-mouse Iggκ-HRP (sc-516102, Santa Cruz Santa Cruz Biotechnology, Dallas, TX, USA) was used as a secondary antibody for Visualization was carried out using Pierce^TM^ ECL reagents (Thermofisher, Waltham, MA, USA).

### 2.9. Proliferation Assay

H1299 cell line was plated in 96 well plates, at density of 7 × 10^3^ cells/well. The next day the cells were infected with TLR5-WT-pAd1 or TLR5-N592S-pAd1 constructs and 2 h post infection the medium was replaced with fresh medium. For the control uninfected condition, the medium was also replaced with fresh medium. After two hours of recovery, the cells were treated with a different concentration of chemotherapeutics, while the concentration of flagellin was held constant (final flagellin concentration was 50 ng/μL). The cells were treated with next chemotherapeutics: paclitaxel (final concentrations 2, 2.5, and 4 nM), carboplatin (final concentrations 150, 170 and 200 μM) and cisplatin (final concentrations 35, 45, and 70 μM). After 72 h, the medium was replaced with 5 mg/mL MTT solution. After 4 h of incubation DMSO was added and the absorbance was measured at 570 nm.

### 2.10. Serum Concentration of the Cytokines

The human serum was separated from peripheral blood of the individuals with a known genotype. Peripheral blood (3 mL) was collected during regular control medical examination and inclusion criteria were that patients should be in the stable state of the disease. Blood samples were centrifuged and serum was separated and stored at −80 °C.

Concentrations of the selected cytokines in the sera of COPD patients and healthy donors were measured using a ProcartaPlex High Sensitivity Assay, with a corresponding bead set (Thermo Fisher Scientific, Waltman, MA, USA), according to manufacturer’s recommendation. Following the detailed protocol which is described here [21], labeled samples were analyzed by use of a Luminex 200 instrument. The concentration of tested cytokines was determined by interpolation from a standard curve using the xPONENT software package (Luminex, Austin, TX, USA).

### 2.11. RNA-Seq Library Preparation and Sequencing

H1299 and WI-38 cell lines were seeded in T-25 flasks, 7 × 10^5^ cells. The next day, the cells were infected with adenovirus 5 containing TLR5^WT^ and TLR5^N592S^ constructs, respectively. The next day, cells were stimulated with flagellin (final concentration was 50 ng/μL). After 24 h of stimulation the cells were harvested and the total RNA was isolated using RNeasy Plus Mini Kit (QIagen, Hilden, Germany) according to the manufacturer’s instructions. The RNA quality was assessed by using Bioanalyzer RNA 6000 Nano Chip (Agilent, Santa Clara, CA, USA). Then, 1 μg of total RNA was used for library preparation using Universal Plus mRNA-Seq Kit (NuGEN, Männedorf, Switzerland) according to the manufacturer’s protocol. The quality of the final libraries was assessed using Bioanalyzer High Sensitivity DNA Chips (Agilent, Santa Clara, CA, USA). The libraries for each condition were prepared in technical triplicates. Paired-end sequencing was performed on Illumina HiSeqX platform, with a read length of 151 bp.

### 2.12. Bioinformatics Analysis

Quality control of raw fastq files was performed using the FastQC program (version 0.11.9). After the files passed the quality control, adapters were trimmed using the fastq program (version 0.20.1) and reads were aligned using STAR (version 2.7.6a) on GRCh38 as a reference genome. Reads were counted using Salmon (version 1.4.0). Differentially expressed genes (DEGs) were obtained using DESeq2 (version 3.12) package in R (version 4.0.3). *p* value was adjusted using Benjamini–Hochberg method [23]. As a cut-off genes with adjusted *p* value (padj) < 0.05 and with log2 fold change (Log2FC) ≥ 1.5 and ≤−1.5 were considered significant. To explore the possible functions of all DEGs, including those that were not considered significant, we performed gene ontology (GO) functional enrichment using GOrilla (https://http://cbl-gorilla.cs.technion.ac.il/, accessed on 13 October 2020) and KEGG (Kyoto Encyclopedia of Genes and Genomes) pathway enrichment analysis using GSEA (version 4.1.0). We considered GO terms and KEGG pathways with adjusted *p* value < 0.05 as statistically significant.

### 2.13. Statistical Analysis

Experimental data were analyzed using GraphPad Prism 6 (GraphPad Software, Inc., San Diego, CA, USA). Statistical significance of the results obtained by the in vitro analysis was determined by the choice of the parametric or non-parametric tests as indicated in the figure legends. Generally, *p*-values < 0.05 were considered as statistically significant and denoted by an asterisk (*), *p*-values < 0.01 were denoted by (**) while *p*-values < 0.001 were denoted by (***).

## 3. Results

### 3.1. TLR5 Variant N592S Is Associated with an Increased Risk for COPD and NSCLC Development

The aim of the genetic association analysis was to investigate the relationship between the frequency of the *TLR5* genotypes and the risk for COPD and lung cancer development. We performed a case–control study and analyzed the genotype frequencies between healthy donors as controls, and clinically defined groups of patients: patients diagnosed with COPD independently of LC status, patients diagnosed with LC of any type independently of COPD status and patients diagnosed with NSCLC independently of COPD status. Secondly, we investigated the association between the frequency of the *TLR5* genotypes and NSCLC development in the group of patients with COPD in the background. The study was carried out for 3 different SNPs located in *TLR5*; 2 nonsynonymous SNPs (rs2072493_N592S_AG and rs574174_F616L_TC) and 1 located in the promotor region of the *TLR5* gene (rs725084_AG). Genetic associations were estimated by logistic regression and adjusted for age, gender, and smoking status. Results of this analysis revealed that only the nonsynonymous *TLR5* SNP, rs2072493, coding for N592S, was statistically significantly associated with the risk of developing COPD, LC, and NSCLC (Table 4).

In this analysis, the presence of the rs2072493_N592S minor allele was associated both with increased risk for COPD development (A/G + G/G OR = 4.41 [2.36–8.24] *p* < 0.0001) and development of lung cancer of any type (A/G + G/G OR = 4.61 [2.15–9.87] *p* = 0.0001). When patients with NSCLC type of cancer were separately analyzed, we detected that the presence of the rs2072493_N592S minor allele, both in co-dominant and dominant model, was statistically significantly associated with the development of NSCLC (A/G OR = 4.93 [2.12–11.49] *p* = 0.0002; G/G OR = 8.44 [1.25–56.85] *p* = 0.0284; A/G + G/G OR = 5.17 [2.37–11.31] *p* < 0.0001).

As already mentioned, COPD is a leading risk factor for NSCLC development, independently of smoking history, and the genetic background of this phenomenon is still elusive. In order to determine if the selected SNPs were associated with NSCLC development in patients with COPD, we compared genotype distribution between the following groups of patients: COPD patients diagnosed with NSCLC (cases) vs. COPD_only_ (controls) (Table 5).

Results of this analysis show that out of 3 tested SNPs, only rs2072493_N592S was associated with NSCLC development in patients co-diagnosed with COPD (A/G OR = 1.52 [1.02–2.25] *p* = 0.0365; G/G OR = 4.49 [1.9–10.61] *p* = 0.0006; A/G + G/G OR = 1.75 [1.21–2.54] *p* = 0.0031).

In conclusion, the genetic association studies performed here indicated that the rs2072493 gene variant, coding for N592S, in *TLR5* gene, could potentially be considered as a genetic biomarker associated with the increased risk for COPD and NSCLC development. The same variant was associated with the increased risk for the NSCLC development in the patients co-diagnosed with COPD.

### 3.2. TLR5 Variants Are Associated with the Lymph Node Involment (N Status) and Overall Survival in NSCLC Patients

The aim of this analysis was to also investigate if any of tested SNPs in TLR5 were associated with clinical characteristics in COPD and lung cancer patients. In this analysis, we included following clinical parameters: forced expiratory volume in 1 s (FEV_1_), disease stage (stage I to IV) and TNM status according to the International Union Against Cancer Criteria (UICC). For the purpose of this analysis Fishers’ test was used and showed that the rs725084 minor allele in dominant model (T/C + C/C) was significantly associated with N status (*p* = 0.0173). Pairwise comparison analysis using Fisher’s exact test showed that the frequency of the minor allele was significantly higher in patients with N2 and N3 tumors when compared to patients with N0 and N1 status (*p* = 0.038). 

Next, we evaluated the effect of the SNPs on survival of the lung cancer patients in order to obtain a better insight how tested *TLR5* SNPs influence the survival rate of the carrier. The results of this analysis are presented in Figure 1 (only associations with statistical significance are presented).

As shown in Figure 1, two out of three tested TLR5 variants were associated with survival in lung cancer patients (rs5744174_F616L and rs725084), as opposed to rs2072493_N592S, which showed no association. Additionally, none of the tested SNPs were associated with survival in the COPD patients. The non-synonymous SNP rs5744174, coding for F616L in TLR5, was associated with overall survival in a group of patients diagnosed with lung cancer of any type (*p* = 0.0205) and with the survival in NSCLC patients (*p* = 0.0056). When NSCLC patients were subdivided into adenocarcinoma (AdC) and squamous cell carcinoma (SQC) subgroups, it showed that rs5744174 was associated only with SQC (*p* = 0.0115). Next, we showed that the presence of rs5744174 minor allele in a co-dominant model was associated with better overall survival in SQC patients (*p* = 0.036; HR 0.4571). Finally, for the TLR5 promoter SNP rs725084 we found an association with NSCLC_AdC patients’ survival (*p* = 0.0279), and with NSCLC_SQC patients’ survival in co-dominant model (*p* = 0.0416; HR 0.2529), indicating the association of minor allele with better overall survival.

### 3.3. N592S Variant Affect the Activation of the NF-κB and AP-1 Transcription Factors

The aim of this analysis was to test if the presence of the rs2072493_N592S mutation, which we found to be associated with COPD and NSCLC development, could impact the activation of two different transcription factors, NF-κB and AP-1. The function of NF-κB and its role in TLR signaling are well recognized—all TLR signaling pathways, including TLR5, culminate in the activation of the NF-κB, which controls the expression of an array of inflammatory cytokine genes. On the other hand, members of the transcription factor activator protein 1 (AP-1) family are known activators of oncogenic transformation and its activation is also potentiated with TLR signaling. Therefore, for the purpose of this analysis, the WI-38 cell line (human fibroblasts isolated from the lung tissue) was transiently transfected with pCDNA3.1_TLR5^WT^ or pCDNA3.1_TLR5^N592S^ plasmids together with luciferase reporter plasmids under the NF-κB or AP-1 control (Figure 2).

Results of this analysis clearly show that the presence of the rs2072493_N592S variant affected the efficiency of the tested transcription factors. We observed that WI-38 cells transfected with rs2072493_N592S gene variant, upon stimulation with 50 ng/mL of flagellin, exhibited significantly lower activation of NF-κB transcription factor, when compared to wild type (*p* = 0.0284). In the same experimental conditions, we observed that AP-1 transcription factor activity, which was found to be high in unstimulated cells, was also affected by the presence of rs2072493_N592S mutation. We detected a statistically significant increase in basal AP-1 transcription factor activity in the presence of the rs2072493_N592S gene variant, relative to the wild-type, in WI-38 cell line, both in endogenous unstimulated conditions (*p* = 0.0013), and stimulated with flagellin (*p* = 0.0179). It is also worthwhile to mention that the same set of experiments was performed in H1299 cell line. However, we were not able to detect NF-kB and AP-1 activation upon flagellin stimulation (data not shown).

### 3.4. Activation of p38 and ERK Is Affected by the N592S TLR5 Coding Variant in the WI-38 Cell Line

To further address the potential mechanisms underlying the impact of rs2072493_N592S gene variant on NF-κB and AP-1 activation we performed immunoblot analysis and examined in more details how important components of the signaling pathways are affected by *TLR5*-coding variant rs2072493_N592S (Figure 3). Immunoblot analyses were performed on WI-38 and H1299 cell lines infected with TLR5^WT^ or mutated TLR5^N592S^ adenoviral constructs and stimulated with flagellin, in different time points.

It is well known that NF-κB activation requires the phosphorylation and degradation of inhibitory kappaB (IκB) proteins triggered by two kinases, IκB kinase alpha (IKKα), and IKKβ. Results of the immunoblot analysis, using antibody against IκB, showed that there is no significant differences in NF-κB activation, measured by IκB degradation, between WT and the N592S TLR5 variant. The results of immunoblot analysis, showing that there is no NF-κB activation in H1299 cell line upon stimulation (Figure 3B; IκB degradation) are important because they confirmed our results obtained by signaling assay. Namely, after stimulation with a specific TLR5 ligand, we were not able to activate TLR5-signaling pathway in H1299 cell line, and to detect NF-κB activation, measured by luciferase assay. Finally, when we analyzed the activation of the MAPKs, p38, and ERK, measured by their phosphorylation status, we observed significant activation shift in WI-38 cells in the 15 and 30 min post-stimulation.

### 3.5. H1299 Cells Overexpressing N592S Variant Exhibit Increased Chemosensitivity

Given the fact that TLR activation, in general, could be associated with cell death by triggering apoptosis, we were interested in how the presence of rs2072493_N592S variant affects the cell response to chemotherapeutic agents currently used in NSCLC treatment. Therefore, we analyzed the induction of cell death, measured as proliferation rate in cells co-stimulated with flagellin and selected agents. For the purpose of this analysis, H1299 cells were infected with adenoviral constructs, TLR5^WT^ or TLR5^N592S^, or left uninfected (N), and co-stimulated with 50 ng/μL of flagellin and increased concentrations of three different chemotherapeutic drugs, as indicated in Figure 4.

First, we determined the IC_50_ concentrations for each drug (45 μM for cisplatine, 90 μM for carboplatine, and 3 nM for paclitaxel for H1299 cell line; data not shown) and used similar concentrations in our experiments. Results of this analysis have shown that H1299 cells, overexpressing TLR5^N592S^, and co-treated with flagellin and increased concentrations of cisplatine, exhibit statistically significant reduction in proliferative rate, in comparison to TLR5^WT^ expressing cells. The same effect was observed for the carboplatine and paclitaxel. These results suggest that tumoral cells overexpressing TLR5^N592S^ variant are more sensitive to chemotherapy induced cell death in the presence of flagellin. In other words, they exhibit increased chemosensitivity after co-stimulation with flagellin and selected chemotherapeutic agents suggesting that TLR5 could play important role in this process.

### 3.6. ELISA

The results of our association studies indicated an association between the rs2072493_N592S coding variant and the risk for COPD development. Additionally, we have found that COPD patients carrying this allele have an increased risk of developing NSCLC. Therefore, we explored whether the serum concentration of the pro-inflammatory cytokines (IL-6, IL-8, IL-1α, IL-1β, and TNFα), down-stream targets of TLR5 activation, could be affected by rs2072493. For the purpose of this analysis COPD subjects were carefully selected, only those in the stable state of the disease were included in the study. The sera were collected from the whole blood. Results of this analysis, presented in Figure 5, show that cytokine sera concentrations between healthy and COPD donors are not dramatically affected by rs2072493_N592S.

### 3.7. Transcriptome Analysis

In order to gain better insights into the consequences of the rs2072493_N592S variant on the transcriptional changes in WI-38 and H1299 cells, they were infected with TLR5-WT-pAd1 and TLR5-N592S-pAd1 adenoviral constructs, stimulated with flagellin for 24 h, and subjected to RNA-seq analysis. For the purpose of this analysis, total RNA was isolated from the treated cells and libraries were constructed from the three independent biological replicates to analyze mRNA. After sequencing, the raw data were processed and analyzed using DESeq2 package in R program. We obtained a list of differentially expressed genes between indicated cell lines. Adjusted *p* value (p_adj_) < 0.05 and with log2 fold change (Log2FC) ≥ 1.5 and ≤−1.5 were considered significantly up-regulated and down-regulated, respectively. The lists of all identified transcripts in cell line overexpressing TLR5^N592S^ in both cell lines are shown in Appendix A. In the Appendix A, we listed only functionally relevant genes involved in the regulation of the important cellular functions associated with cancer development, in the first place regulation of the immune response, cell proliferation, and apoptosis, including brief description of their function.

The volcano plot of differentially expressed transcripts in WI-38 and H1299 cell lines was constructed in order to indicate the general scattering of the transcripts and to filter the differentially expressed transcripts for the indicated groups of experimental conditions (WI-38 and H1299 cells infected with TLR5-WT-pAd1 and TLR5-N592S-pAd1). Results of this analysis are shown in Figure 6.

The application of the DESeq2 analysis identified 6 differentially expressed genes in H1299 cell line (log2-fold change ≥ 1.5 and ≤−1.5, adjusted *p*-value < 0.05), 3 of which were up-regulated and 3 down-regulated. By applying the same methodology, in the Wi-38 cell line, we identified 25 differentially expressed genes, 18 of which were up-regulated and 7 down-regulated (Appendix A). Comparing the results presented in Appendix A, it is evident that there are no common differentially expressed genes between the two cell lines. However, we observed that differentially up-regulated/down-regulated genes, in the both cell lines, shared similar functional annotations which are essential for: the regulation of tissue homeostasis; regulation of inflammatory response (CHRFAM7A; log2Fc = 2.76; *p* = 0.028); regulation of cytokine production (C1QTNF3; log2Fc = 7,78; *p* = 0.00081); transcription regulation, DNA repair, DNA replication, and chromosomal stability (H2AC19; log2Fc = 3.38, *p* = 1.85 × 10^−0.6^); and modulation of autophagy processes and dendritic cell activation (LAMP3; log2Fc = 3.83, *p* = 0.04).

To further understand the functional/biological consequences of the rs2072493_N592S variant we sought to identify transcriptomic pathways affected by N592S overexpression. We performed a gene ontology analysis on a set of differentially expressed genes and determine which cell components, functions, and processes are affected in H1299 and WI-38 cell lines overexpressing TLR5-WT or TLR5-N592S. The input was a list of up- and down-regulated genes, ranked according to log2FC. Statistically significant GO terms were those whose false discovery rate q value (FDR q value), which adjusts the *p* value for multiple testing, was <0.05. The results are shown in Figure 7.

Although there are no statistically significant genes that overlap in the analysis of differentially expressed genes between H1299 and WI-38 cell lines, here we have several GO terms that are in common for both lines of cells. The common GO terms affecting the cellular components which were significantly enriched were mostly associated with cellular membrane (cell projection membrane, receptor complex and parts of the plasma membrane). The common GO terms affecting the cellular function were mostly involved in the regulation of the receptor and ligand activity and molecular transducer activity. Finally, the common GO terms affecting the biological cell process were mostly involved in G protein-coupled receptor signaling pathway.

Finally, we performed Gene Set Enrichment (GSE) analysis to identify which pathways that are in the KEGG database are enriched. Results of this analysis are listed in Appendix A. and demonstrated in Figure 8.

Gene set enrichment analysis identified multiple pathways that were significantly down- or up-regulated by overexpression of TLR5^N592S^ overexpression in WI-38 and H1299 cell lines. Among them, we found that pathways participating in the antigen processing and presentation (FDR q value = 0.025194; enrich. score −1.4960678) and NK-cell mediated cytotoxicity (FDR q value = 0.042227; enrich. score −1.3609923) exhibit significant negative enrichment in H1299 cell line. There were no pathways with a positive enrichment score in GSEA for the H1299 cell line. For WI-38 cell line, we detected that calcium signaling pathway exhibits a significant negative enrichment score, while for tyrosine metabolism pathway we identified positive enrichment score (Appendix A).

In conclusion, we would like to point out several interesting results of the transcriptome analysis. First, we confirmed that the presence of rs2072493_N592S coding variant affected expression profile in both of the tested cell lines, healthy lung fibroblasts, and lung metastasis cells. Second, among differentially expressed genes, PRR7, involved in positive regulation of apoptotic process is up-regulated in the WI-38 cells, and LAMP3, regulator of dendritic cell activation is down-regulated, indicating that rs2072493_N592S affecting important process in carcinogenesis. Finally, results of GSEA analysis in H1299 cells identified specific pathways regulating the antigen processing and presentation, and cellular mediated cytotoxicity to be significantly down-regulated.

## 4. Discussion

Chronic inflammation, incidence of infections, and risk of developing the lung cancer have recently emerged as increased in patients with COPD, suggesting that altered immune response can jeopardize innate protective mechanisms for malignancy. Genetic mapping has detected several SNPs underlying COPD and lung cancer, most of which belong to different gene families, such as proteinases and inflammatory cytokines [24]. Coding, non-synonymous SNPs may result in amino acid substitutions directly altering the protein itself and affecting protein functions, such as the ability of the receptor to bind pathogens. Alternatively, they may lead to deficiencies in intracellular transport or changed interaction with the adaptive proteins [25]. Furthermore, non-coding SNPs may also alter gene regulation by modulating promotor activity, splicing, or mRNA stability, resulting in differential expression. If such functional SNPs have been ascribed notable functional repercussions in a specific disease, e.g., another type of cancer, it can be presumed that such predisposition loci have overlapping effects and may be relevant for another disease entity. In the presented study, we analyzed three SNPs located in *TLR5* due to the reported relevance of *TLR5* SNPs in colorectal cancer [19], obesity, and diabetes [26]. We hypothesized that these variants could be related to the development of COPD and/or lung cancer pathogenesis. Here, we found strong evidence of association between the coding *TLR5* SNP rs2072493_N592S and increased risk for developing COPD and NSCLC. Additionally, we observed association of the rs2072493_N592S minor allele with a tendency of NSCLC development in the patients diagnosed with COPD. Our study showed that the presence of the *TLR5* rs2072493_N592S minor allele genotypes G/G + A/G, after age and gender adjustments, could be considered as a genetic risk factor for NSCLC development in COPD patients. We found this observation very interesting, because common mechanisms leading to NSCLC development, when the COPD is in the background, and genetics of this processes are still elusive. In addition to tobacco smoke, which is a common characteristic for both diseases, it is very likely that many other biological processes, such as dysregulated immune response/inflammation, abnormal tissue repair, or cell proliferation, are involved in the pathogenesis of this conditions. Therefore, we assume that rs2072493_N592S variant alter the TLR5 signaling pathway resulting in dysfunctional biological processes which, in the end, culminate with the increased risk of lung cancer development. Results of our in vitro analysis in the lung cancer cell line (H1299) and healthy fibroblasts (WI-38) indicated that the tested variant is associated with lower activity of NF-κB signaling, an important cancer signaling pathway that plays a crucial role in the induction of inflammatory response in lung cancer [27]. In addition, we also observed that IL-6 serum levels were lower in wild type allele carriers (cf. Figure 5). Interleukin IL-6, produced by T and B lymphocytes, phagocytic cells, endothelial cells and, it is worthwhile to mention, airway epithelial cells, together with many other cells, are important regulators of the immune response and inflammation. IL-6 production is mainly operated via NF-kB transcription factor and its secretion in airway epithelia is induced by flagellin, a compound of bacterial flagellae, a strong mediator of pulmonary inflammation, and cognate TLR5 ligand. Ritter et al. showed that a different type of cytokines, including IL-6, are also able to influence the regulation of TLR mRNA and protein expression [28]. However, the increased risk of developing COPD and NSCLC, observed here for rs2072493_N592S, is in slight contrast with lower IL-6 levels, since this cytokine, at least in colorectal cancer, can also have a tumor-promoting role [29] and, hence, IL-6 could be expected to be higher in rs2072493_N592S carriers. On the other hand, earlier findings in HEK293T and healthy donor primary peripheral blood mononuclear cells (PBMCs) implied that rs2072493_N592S seemed to rather cause a hyperresponsive phenotype. The observed higher basal AP-1 activity (cf. Figure 2) aligns with this and the remaining differences may be explained by the differences in cell type and stimulation conditions. Nevertheless, our results clearly show that the rs2072493_N592S variant alters TLR5 function relative to wild type TLR5. Our data warrant a further analysis of this aspect as inflammatory responses play dual role in lung cancer development. Excessive TLR activation can result in uncontrolled inflammatory processes and consequent pulmonary tissue damage. On the other hand, reduced TLR expression and function can lead to an immunosuppressed state. It was proposed by Kutikhin AG et al. that the described scenario could be based on the weakening of immune responses to bacterial or viral agents that increase the risk of infection and disturbed pro-inflammatory cytokine production due to certain molecular changes in TLR pathways [30]. What is most worthy of mention, and had not been investigated before, is the strong effect on the response of lung cancer cell line (H1299) to the combined treatment with frequently used cytostatic drugs and the flagellin, the TLR5 agonist (cf. Figure 4). Here, we observed that the overexpression of the TLR5 rs2072493_N592S allele showed a significant reduction in cell proliferation (i.e., increased chemosensitivity), suggesting individuals carrying this allele may have better treatment responses. We found this as an intriguing result that may contribute to the better understanding of biological processes associated with susceptibility to cancer development, activation of the immune system, and the outcome of the disease. The human lung has its own low-density microbiota and it is now widely accepted that respiratory microbiome, like those in the gut, plays an important role in health and diseases [31]. It is also well accepted that prophylactic antibiotics are commonly used for cancer patients undergoing chemotherapy, in order to reduce the risk of neutropenia-associated infection [32], and we know that the major obstacle in achieving effective antitumor therapy is the immunosuppressive environment generated by the tumor, pointing out the importance of shifting from the immunosuppressive microenvironment toward induction of the innate immune response and cell apoptosis, processes that are all regulated by TLRs [33]. Furthermore, it has been shown in mouse model of cancers that TLR5 ligand, flagellin, inhibited cell proliferation and elicited potential antitumor activity [34]. However, what we do not know is the genetic background of the abovementioned findings. Therefore, we would like to emphasize that our results indicate that it may be informative to investigate specific links between TLR5 genotype and response to therapy with microbiota analysis, especially in the abundance of flagellated bacteria. In order to better understand the functional relevance of the observed chemosensitivity effect, we conducted transcriptomic analysis on RNA isolated from H1299 and WI-38 cell lines overexpressing TLR5 wild type or TLR5 rs2072493_N592S gene variant. By applying transcriptomic analysis we identified several differentially expressed genes important in controlling the biological processes related to cancer development and progression (cf. Appendix A). For example, ZBTB12 is the zinc finger and BTB domain containing protein 12, a predicted transcription factor belonging to the family of methyl-CpG-binding proteins, playing an important role in cell differentiation and malignant transformation [35]. In our study, ZBTB12 exhibits the highest upregulation in the H1299 cell line with strong statistical significance (*p* = 8.76 × 10^−8^). Interestingly, we also observed that H2AC19, a member of the histone H2A family which plays a central role in transcription regulation, DNA repair, DNA replication, and chromosomal stability is significantly up-regulated in WI-38 cell line (*p* = 1.85 × 10^−6^). Results published by Groysman et al. show that H2AC19 gene is induced by chemotherapy and have prognostic value in colorectal carcinoma. In this study, authors investigated the effect of clinical chemotherapeutics on cytokine production profile, that could either promote cancer or have an anti-cancer effect [36]. Bearing in mind that there is individual heterogeneity in response to chemotherapy, not only in lung cancer, but, in general, pan-cancer heterogeneity, our results seems, to us, even more interesting. Namely, in the presented study, we identified rare TLR5 genetic variant strongly associated with COPD and NSCLC development and its overexpression affect response to clinical chemotherapeutics. However, it is also important to emphasize that our research is conducted in in vitro models and further confirmation in clinical setup in needed. Furthermore, it is important to add that the in vitro models (cell lines) we used for the purpose of this study exhibit endogenous expression of TLR5 and it is possible that affected signaling and chemosensitivity observed here could be attributed to the dominant-negative effect of the mutated allele. For final confirmation of this claim, it is necessary to carry out additional in vitro analyzes on cell models with attenuated TLR5 expression, which was out of the scope of this study. In conclusion, our results suggest that *TLR5* could potentially be recognized as a biomarker, not only for COPD and NSCLC development, but also for therapy response, which certainly should be further investigated.

## 5. Conclusions

In the presented study, we performed case–control genetic association and functional studies on the importance of *TLR5* in COPD and LC development. The results of our analysis indicated that the TLR5^N592S^ gene variant is associated with an increased risk of COPD and NSCLC development, and the development of the NSCLC when COPD is in the background. Functional analysis indicated that overexpression of the N592S allele affected the activation of NF-κB and AP-1, and most importantly, with increased chemosensitivity in the H1299 cell line. In conclusion, we can say that our results suggest that *TLR5* could be potentially considered as a biomarker for COPD and LC development with functional relevance, which is reflected in increased sensitivity to chemotherapeutic drugs frequently applied in lung cancer treatment. However, it is important to say that this study is conducted in in vitro models and for stronger confirmation of our results, it is necessary confirm them on clinical materials.

## Figures and Tables

**Figure 1 biomedicines-10-02240-f001:**
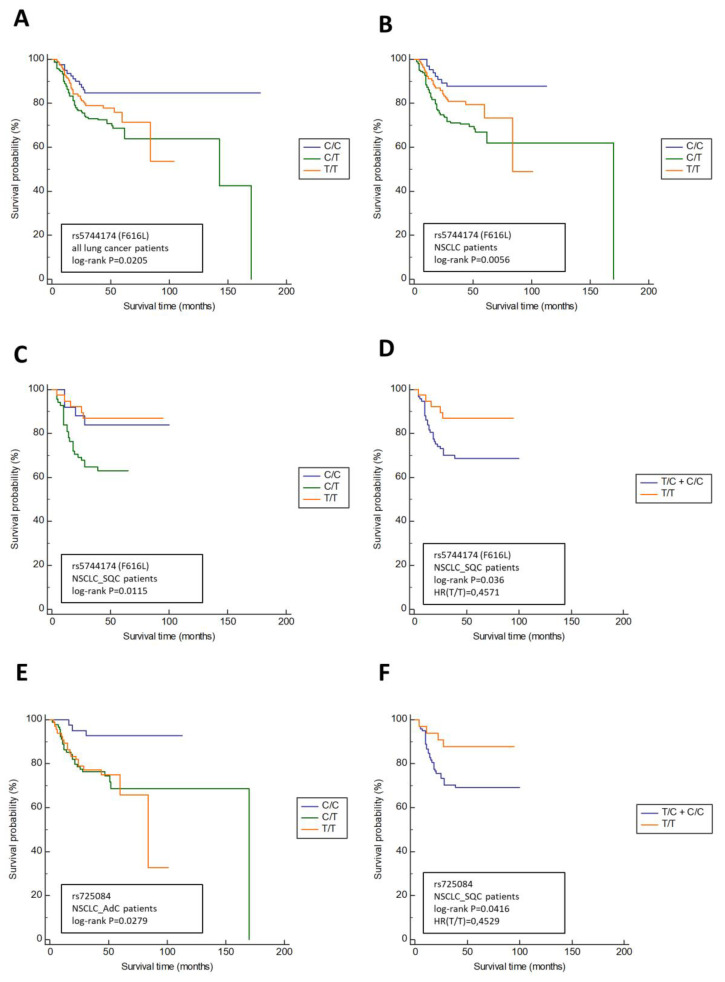
Kaplan–Meier estimations of lung cancer-specific overall survival according to TLR5 genotypes. (**A**) rs5744174_F616L, all lung cancer patients (*n* = 415); (**B**) rs5744174_F616L, only non-small cell lung cancer (NSCLC) patients (*n* = 343); (**C**) rs5744174_F616L, only squamous-cell carcinoma (SQC) patients (*n* = 131); (**D**) rs5744174_F616L in co-dominant model, only SQC patients (*n* = 131); (**E**) rs725084, only adenocarcinomas (AdC) patients (*n* = 197); and (**F**) rs725084 in co-dominant model, only SQC patients (*n* = 131).

**Figure 2 biomedicines-10-02240-f002:**
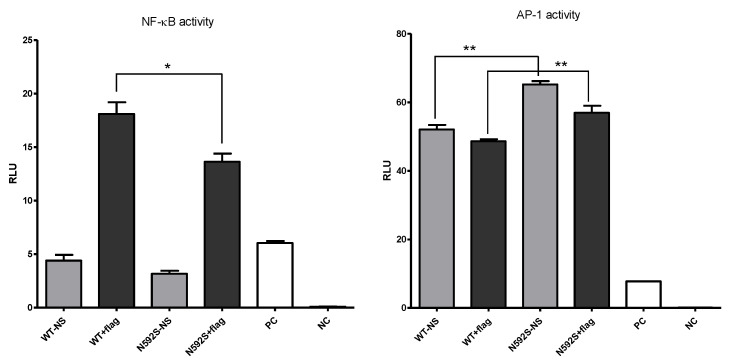
NF-κB and AP-1 transcription factors activity in WI-38 cell lines transfected with WT and N592S TLR5-variant that are stimulated with flagellin (flag) or unstimulated (NS). Positive control (PC) is a mixture of a constitutively expressing GFP construct, constitutively expressing Firefly luciferase construct, and constitutively expressing Renilla luciferase construct, while negative control (NC) is a mixture of non-inducible firefly luciferase reporter and constitutively expressing Renilla construct. Data shown are representative of three independent experiments. Statistical analysis was performed by using GraphPad, *t*-test; *p*-values < 0.05 (*), *p*-values < 0.01 (**).

**Figure 3 biomedicines-10-02240-f003:**
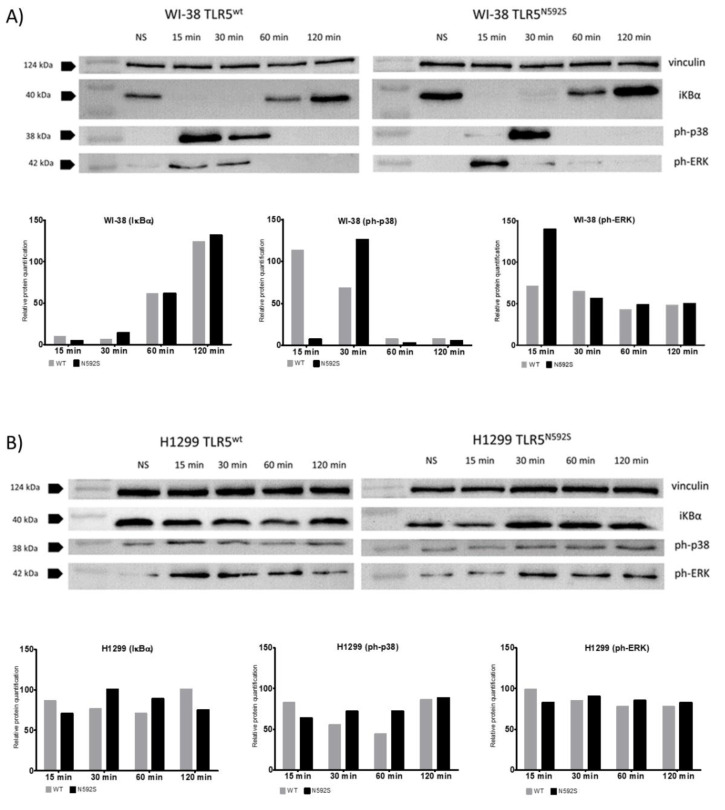
Immunoblot analysis of expression of NF-κB and AP-1 signaling pathways members: pERK, p-p38 and IKBα on (**A**) WI-38 cells infected with TLR5^WT^ and TLR5^N592S^ adenoviral constructs and (**B**) H1299 cells infected with TLR5^N592S^ and TLR5^N592S^ adenoviral constructs. Cells were infected with an adenoviral vector carrying either TLR5^WT^ or TLR5^N592S^ gene and stimulated with a flagellin (50 ng/mL) at five different time points (15, 30, 60, and 120 min). Vinculin was used as endogenous control and for normalization. Biological replicates were made and data shown are representative of two independent experiments.

**Figure 4 biomedicines-10-02240-f004:**
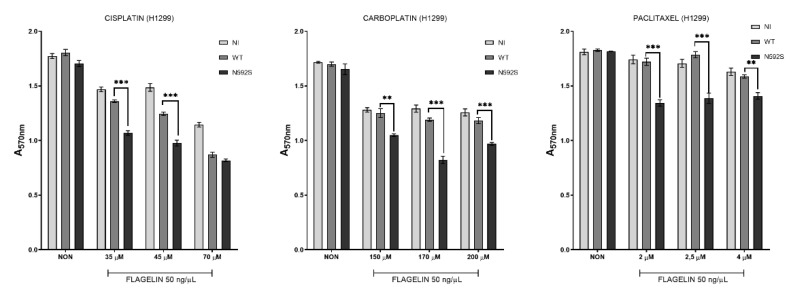
Proliferation analysis of the non-infected (NI) H1299 cells, infected with adenoviral vector carrying wild type of the TLR5^WT^ (WT) or mutated TLR5^N592S^ (N592S) gene. Cells were stimulated with flagelin (50 ng/μL) and increased concentration of different cytostatic, cisplatin (35, 45 and 70 μM), carboplatin (150, 170, and 200 μM) and paclitaxel (2, 2.5, and 4 nM), or left untreated (NON). Statistical analysis was performed by using GraphPad; two-way ANOVA test; ** *p* < 0.01 and *** *p* < 0.001.

**Figure 5 biomedicines-10-02240-f005:**
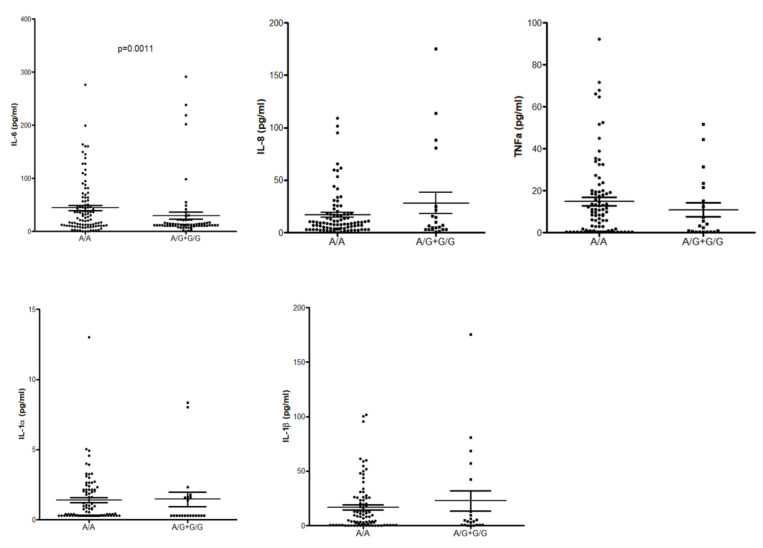
Effect of the TLR5 rs2072493_G/A (N592S) genotypes on serum IL-6, IL-8, TNFα, IL-1α, and IL-1β concentrations. Analysis was performed with a ProcartaPlex High Sensitivity Assay, according to manufactures protocol, which can be found in the Materials and Methods section. Statistical analysis was performed by using GraphPad; non-parametric *t*-test (Mann–Whitney test). Dots represent A/A genotype, squares represent A/G + G/G genotype.

**Figure 6 biomedicines-10-02240-f006:**
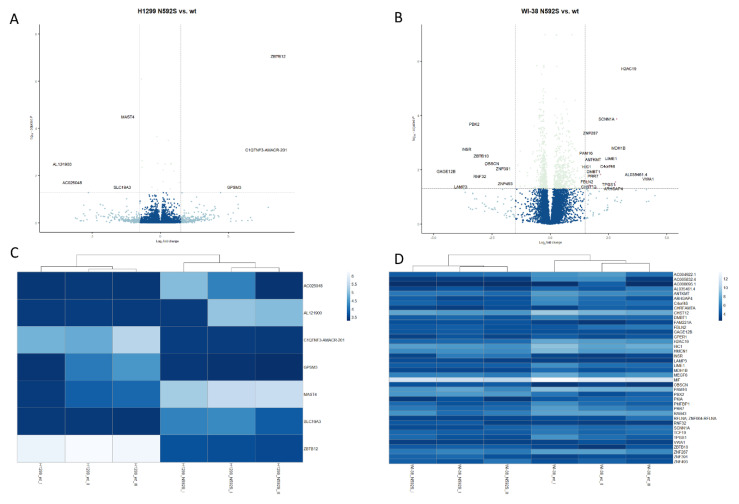
Comprehensive transcriptome analysis of the H1299 and WI-38 cell lines overexpressing TLR5^WT^ or TLR5^N592S^ gene variants. Volcano plot of differentially expressed transcripts in H122 (**A**) and WI-38 (**B**) cell line. Heat map describing expression of statistically significant DEGs with dendograms indicating hierarchical clustering between H1299 (**C**) and WI-38 (**D**) cell lines.

**Figure 7 biomedicines-10-02240-f007:**
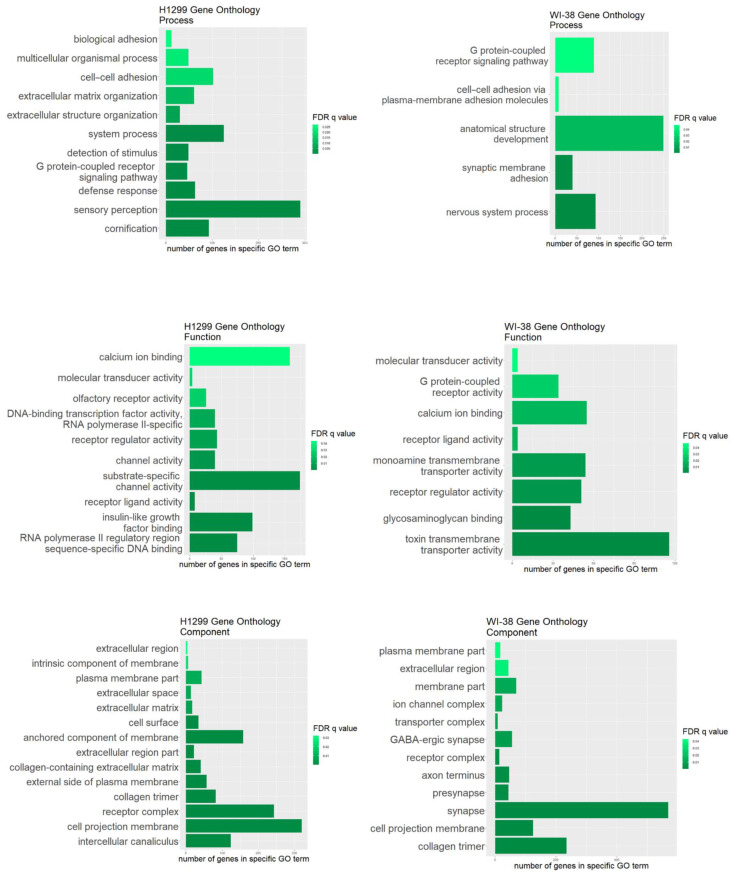
Results of the gene ontology analysis of the differentially expressed genes in H1299 and WI-38 cell lines overexpressing TLR5^N592S^ variant. Statistically enriched process, function and component GO terms are shown and ranked based on the false discovery q-value.

**Figure 8 biomedicines-10-02240-f008:**
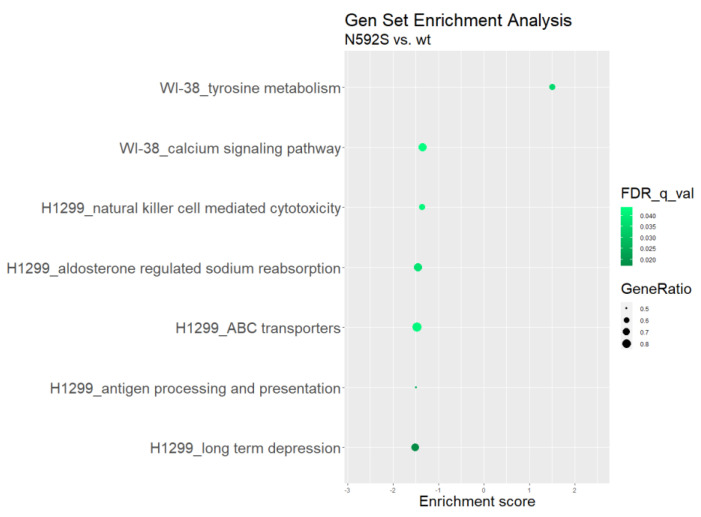
Gen set enrichment analysis showing the positive or negative effect of TLR5 N592S variant on gene set enrichment in H1299 and WI-38 cell line. Size of the dots represents ratio of DEGs in comparison to total number of genes in a given set.

**Table 1 biomedicines-10-02240-t001:** Demographic characteristics of the tested population.

Subjects (N, %)	Cases (974)	COPD (500; 51.3%)	COPD + LC (280; 28.7%)	LC (194; 19.9%)	Controls (1283)
Age, year (SD ^1^)	60 (9.58)	59 (10.4)	63 (7.38)	62 (9.8)	38 (12)
Sex (N, %)					
male	671 (68.9)	355 (71)	205 (73)	111 (57.3)	1120 (87)
female	303 (31.1)	145 (29)	75 (26)	83 (42.7)	163 (12.7)
Smoking status (N, %)					
active	523 (50.8)	139 (25.8)	148 (52.5)	53 (26.3)	325 (25.3)
ex-smoker	347 (33.6)	317 (57.4)	124 (43.9)	90 (43.8)	4 (0.2)
non	104 (9.8)	44 (8)	8 (2.5)	51 (25.3)	954 (74.4)

^1^ Standard Deviation.

**Table 2 biomedicines-10-02240-t002:** Clinical characteristics of the tested population.

CHARACTERISTICS	COPD (N = 500)	COPD + LC and LC_only_ (N = 474)
FEV_1_ % (N) ^1^		
80–120%	55	235
50–80%	207	185
30–50%	148	42
≤30%	90	12
TYPE OF CANCER		
Lung cancer	/	474
NSCLC/AdC	/	225
NSCLC/SQC	/	168
other	/	81
TNM-stage (N)		
1	/	118
2	/	100
3	/	102
4	/	77
nonclasified	/	77
T stage (N)		
1	/	60
2	/	149
3	/	58
4	/	25
nonclasified	/	182
N stage (N)		
0	/	153
1	/	55
2	/	109
3	/	8
nonclasified	/	149
M stage (N)		
0	/	258
1	/	60
nonclasified	/	156

^1^ Forced expiratory volume in 1 s (FEV_1_).

**Table 3 biomedicines-10-02240-t003:** List of the single nucleotide polymorphisms (SNPs) analyzed in the study, minor allele frequency and functional consequence according to SIFT and PolyPhen, respectively.

SNP	Chr.	Position	Allele	Location	MAF ^1^	Aa Change	Sift ^2^	PolyPhen ^3^
rs5744174	1	223284528	C/T	Exon	0.45 (T)	F616L	Tolerated	Benign
rs2072493	1	223284599	A/G	Exon	0.12 (G)	N592S	Tolerated	Benign
rs725084	1	223323951	T/C	promotor	0.44 (T)	/	/	/

^1^ MAF, minor allele frequency; ^2^ SIFT, Sorting Intolerant from Tolerant; ^3^ PolyPhen, Polymorphism Phenotyping.

**Table 4 biomedicines-10-02240-t004:** Association between selected TLR5 SNPs and risk for chronic obstructive pulmonary disease (COPD) and lung cancer (LC) development. COPD cases were selected independently on lung cancer status and same was performed for lung cancer cases—they were selected independently for COPD status.

		COPD vs. Healthy				Lung Cancer vs. Healthy				NSCLC vs. Healthy			
SNP	Genotype	Cases (N, %)	Controls	OR (95% CI)	*p* Value	Cases (N, %)	Controls (N, %)	OR (95% CI)	*p* Value	Cases (N, %)	Controls (N, %)	OR (95% CI)	*p* Value
rs5744174_TC	T/T	216 (33.4)	381 (30.8)	1		145 (33.2)	381 (30.8)	1		132 (32.5)	381 (30.8)	1	
(coding F616L)	T/C	296 (45.7)	606 (49)	0.64 (0.32–1.28)	0.206	210 (48)	606 (49)	1.09 (0.54–2.18)	0.8073	199 (49)	606 (49)	1.33 (0.64–2.74)	0.4384
	C/C	135 (20.9)	249 (20.2)	0.97 (0.47–2.01)	0.9382	82 (18.8)	249 (20.2)	0.38 (0.13–1.14)	0.0851	75 (18.5)	249 (20.2)	0.46 (0.15–1.43)	0.1803
	T/C + C/C	431 (66.6)	855 (69.2)	0.87 (0.51–1.51)	0.6314	292 (66.8)	855 (69.2)	0.88 (0.45–1.71)	0.6987	274 (67.5)	855 (69.2)	0.96 (0.56–1.64)	0.8854
rs725084_TC	T/T	208 (32.4)	325 (26.6)	1		124 (28.8)	325 (26.6)	1		111 (28.1)	325 (26.6)	1	
(promoter)	T/C	301 (46.9)	628 (51.4)	0.85 (0.62–1.16)	0.3126	217 (50)	628 (51.4)	1.12 (0.78–1.59)	0.5431	201 (50.7)	628 (51.4)	1.07 (0.72–1.57)	0.744
	C/C	133 (20.7)	269 (22)	0.83 (0.57–1.22)	0.3502	91 (21.1)	269 (22)	0.81 (0.52–1.26)	0.3496	84 (21.2)	269 (22)	0.82 (0.51–1.33)	0.4311
	T/C + C/C	434 (67.6)	897 (73.4)	0.85 (0.63–1.13)	0.2635	308 (71.1)	897 (73.4)	1.02 (0.72–1.43)	0.9213	285 (71.9)	897 (73.4)	0.99 (0.68–1.43)	0.95
rs2072493_AG	A/A	413 (64.5)	967 (80)	1		248 (57.7)	967 (80)	1		226 (57.1)	967 (80)	1	
(coding N592S)	A/G	196 (30.6)	226 (18.7)	4.48 (2.34–8.57)	**<0.0001**	150 (34.9)	226 (18.7)	4.63 (2.06–10.39)	**0.0002**	142 (35.8)	226 (18.7)	4.93 (2.12–11.49)	**0.0002**
	G/G	31 (4.9)	15 (1.3)	4.08 (0.72–23.13)	0.1122	32 (7.4)	15 (1.3)	3.94 (0.77–20.14)	0.0995	28 (7.1)	15 (1.3)	8.44 (1.25–56.85)	**0.0284**
	A/G + G/G	227 (35.5)	241 (20)	4.41 (2.36–8.24)	**<0.0001**	182 (42.9)	241 (20)	4.61 (2.15–9.87)	**0.0001**	170 (42.9)	241 (20)	5.17 (2.37–11.31)	**<0.0001**

**Table 5 biomedicines-10-02240-t005:** Association of selected *TLR5* SNPs with non-small cell lung cancer (NSCLC) development in the cohort of patients co-diagnosed with COPD.

		NSCLC + COPD (Cases) vs. COPD (Controls)			
SNP	Genotype	Cases (N, %)	Controls (N, %)	OR (95% CI)	*p* Value
rs5744174_TC	T/T	43 (17.7)	88 (22.6)	1	
(coding F616L)	C/T	118 (48.6)	173 (44.3)	1.41 (0.87–2.28)	0.1558
	C/C	82 (33.7)	129 (33.1)	1.43 (0.86–2.36)	0.1594
	C/T + C/C	200 (82.3)	302 (77.4)	1.41 (0.90–2.19)	0.1312
rs725084_TC	T/T	45 (18.7)	84 (21.6)	1	
(promoter)	C/T	121 (50.4)	174 (44.8)	1.18 (0.73–1.91)	0.4912
	C/C	74 (30.9)	130 (33.6)	1.06 (0.64–1.73)	0.8156
	C/T + T/T	195 (81.3)	304 (78.4)	1.13 (0.73–1.76)	0.5683
rs2072493_AG	A/A	144 (59.5)	264 (68.6)	1	
(coding N592S)	A/G	80 (33.1)	110 (28.6)	1.52 (1.02–2.25)	**0.0365**
	G/G	18 (7.4)	11 (2.8)	4.49 (1.9–10.61)	**0.0006**
	A/G + G/G	98 (40.5)	121 (31.4)	1.75 (1.21–2.54)	**0.0031**

## Data Availability

Not applicable.

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
