# Peer review of "TLR5 Variants Are Associated with the Risk for COPD and NSCLC Development, Better Overall Survival of the NSCLC Patients and Increased Chemosensitivity in the H1299 Cell Line"

_biomedicines, 2022, doi:10.3390/biomedicines10092240_

Round 1
Reviewer 1 Report
The authors performed a case-control genetic association study on the TLR5 variants in chronic obstructive pulmonary disease (COPD) and lung cancer development and they identified that TLR5 SNP N592S is associated with increased risk for the COPD and non-small-cell lung cancer (NSCLC) development, and NSCLC risk in the COPD patients. The in vitro functional study showed that overexpression of the N592S allele affected the activation of NF-κB and AP-1 pathways in WI-38 cell line and was associated with increased chemosensitivity in the human NSCLC cell line H1299. The RNAseq results showed that several cellular pathways potentially associated with a dysregulated immune response were affected in TLR5-N592S overexpressing cell lines. Although these results are interesting and provide new insights into the TLR5 SNPs and COPD and NSCLC development, there are several issues that are needed to be addressed in the manuscript.
1. Fig. 2: pcDNA3.1_empty vector should be used as an experimental control. More detailed information should be provided in the figure legend for “NS”, “NC”, etc.
2. Fig. 2 and Fig. 3: The expression of pcDNA3.1_TLR5-WT and TLR5-N592S should be confirmed by western blot. How is the endogenous TLR5 level in WI-38 cell lines and H1299 cell line? Overexpression of pcDNA3.1_TLR5 N592S may interfere with the function of endogenous TLR5 in the transferred cell lines, so it is hard to evaluate if the results were due to the transferred TLR5 N592S by itself, or the transferred TLR5N592S competing the binding sites with endogenous TLR5, which may be leading to the loss of function of endogenous TLR5. To avoid the endogenous TLR5 expression interfering with the assay, I suggest the authors use a TLR5 null cell line to perform this experiment, or they can knockdown the endogenous TLR5 before the transfection.
3. Fig. 3: Did the authors check earlier time points for H1299 cell line, for example, 5 min or 10 min? The kinetic of phosphorylation of ERK, p38 in H1299 cell line may be different from that in WI-38 cell lines.
4. Fig. 4. (1) The authors mentioned that activation of p38 and ERK (important in cell proliferation) in cells overexpressing N592S variant may interfere with the responses to chemotherapeutics, however, there is no NF- κB activation in H1299 cell line upon stimulation. Therefore, the increased chemosensitivity for H1299 cells overexpressing TLR5N592S may not be due to the activation of p38 and ERK. The authors should discuss this. (2) Did the overexpression of TLR5-N592S in WI-38 cells exhibit increased cell proliferation?
5. Results 3.7. Transcriptome analysis: How can the authors use very limited number of differential expression genes (only 3 up- and 3 down-regulated) for GO and GSE analysis? I think it is too strict if the authors used log2-fold change >=1.5 and <=-1.5 for their RNA-seq analysis. If the authors only want a 1.5-fold change of expression level included in the analysis, the authors can use log2-fold change >=0.58 or <=-0.58, so the authors can get more genes for bioinformatics analysis.
6. Table 6 takes too much space in the manuscript and should be provided in the supplemental file.
Minor points:
1. Many typos were found in the manuscript. For examples,
Line 404: “indicateing” should be “indicating”; “overall” should be “overall”;
Line 500: “carefuly” should be “carefully”.
Line 501: “colected” should be “collected”;
Line 503: “dramaticaly” should be “dramatically”;
Line 592: “Gen set” should be “Gene set”;
Line 704: “affect” should be “affected”;
Line 722: “confirm” should be “to confirm”.
2. I suggest that the authors use either rs2072493 or N592S or rs2072493/N592S together throughout the manuscript to avoid confusing to the readers.
3. Most sentences in the first paragraph should be moved to Materials and Methods.
4. The font and font size need to be consistent in the manuscript.
5. Most of the “%” in Table 4 can be deleted.
Reviewer 2 Report
This paper looks at the important relationship between inflammation and chronic disease such as COPD and cancer.
Using case-control design they were able to identify the coding variant TLR5 SNP rs2072493 as associated with increased risk for developing COPD and NSCLC. Using cell cultures, the effect of TLR5 gene variants on p38 and ERK activation were studied as well as transcriptome.
Authors suggest that TLR5 could be considered as biomarker for COPD and LC
The underlying data material for the case-control study is adequate and the analytic technology seem appropriate.
I am not totally convinced that TLR5 variant could serve as a biomarker, but I do think that the present study may lead to a better understanding and lead to further research in the role of flagellin and TLR’s in the tumour microenvironment
The paper includes a panorama of various designs. Taken together they provide a very comprehensive study, but the flip side is that the paper appears very extremely data and information dense.
I have no specific comments or suggestions as the paper is well written and the authors clearly wants the reader to understand the various elements.
My suggestion to the authors is to consider if the paper could be divided into to papers and then to seek simultaneous publication. Alternatively, the authors should consider moving some of the material to a appendix such as some of tables (6 and 7)
Author Response
- ANSVER TO THE REWIEVER #2 COMMENTS:
This paper looks at the important relationship between inflammation and chronic disease such as COPD and cancer. Using case-control design they were able to identify the coding variant TLR5 SNP rs2072493 as associated with increased risk for developing COPD and NSCLC. Using cell cultures, the effect of TLR5 gene variants on p38 and ERK activation were studied as well as transcriptome. Authors suggest that TLR5 could be considered as biomarker for COPD and LC. The underlying data material for the case-control study is adequate and the analytic technology seems appropriate.
(1) I am not totally convinced that TLR5 variant could serve as a biomarker, but I do think that the present study may lead to a better understanding and lead to further research in the role of flagellin and TLR’s in the tumour microenvironment.
We understand Reviewers comment that our statement on TLR5 as a biomarker could be considered as to strong. Therefore we accepted Reviewers suggestion and change it accordingly throughout the entire manuscript.
(2) The paper includes a panorama of various designs. Taken together they provide a very comprehensive study, but the flip side is that the paper appears very extremely data and information dense. I have no specific comments or suggestions as the paper is well written and the authors clearly want the reader to understand the various elements. My suggestion to the authors is to consider if the paper could be divided into two papers and then to seek simultaneous publication.
We understand Reviewers comments and concerns regarding the amount of data presented in the manuscript. We agree that it can be separated in two papers, however our idea was to present whole workflow, from associations to the functional studies. Therefore we would like to keep the current format. We hope that our decision will be satisfactory for the Reviewer.
(3) Alternatively, the authors should consider moving some of the material to an appendix such as some of tables (6 and 7).
We accepted Reviewers comment and moved Tables 6 and 7 to the Supplementary Tables 2 and 3.
Round 2
Reviewer 1 Report
The authors have largely addressed the detailed questions I raised in my previous review.